# Semantic Retrieval Augmented Contrastive Learning for Sequential Recommendation

**Ziqiang Cui[1][*], Yunpeng Weng[2][3][*], Xing Tang[4][†], Xiaokun Zhang[1], Shiwei Li[2], Peiyang Liu[5], Bowei He[1], Dugang Liu[6], Weihong Luo[3], Xiuqiang He[4], Chen Ma[1][†],**

[1] City University of Hong Kong    [2] Huazhong University of Science and Technology
[3] Tencent    [4] Shenzhen Technology University    [5] Peking University    [6] Shenzhen University
ziqiang.cui@my.cityu.edu.hk, {wengyp, lishiwei}@hust.edu.cn, xing.tang@hotmail.com,
{dawnkun1993, dugang.ldg}@gmail.com, liupeiyang@pku.edu.cn, boweihe2-c@my.cityu.edu.hk,
lobby66@163.com, hexiuqiang@sztu.edu.cn, chenma@cityu.edu.hk

## Abstract

Contrastive learning has shown effectiveness in improving sequential recommendation models. However, existing methods still face challenges in generating high-quality contrastive pairs: they either rely on random perturbations that corrupt user preference patterns or depend on sparse collaborative data that generates unreliable contrastive pairs. Furthermore, existing approaches typically require predefined selection rules that impose strong assumptions, limiting the model's ability to autonomously learn optimal contrastive pairs. To address these limitations, we propose a novel approach named Semantic Retrieval Augmented Contrastive Learning (SRA-CL). SRA-CL leverages the semantic understanding and reasoning capabilities of LLMs to generate expressive embeddings that capture both user preferences and item characteristics. These semantic embeddings enable the construction of candidate pools for inter-user and intra-user contrastive learning through semantic-based retrieval. To further enhance the quality of the contrastive samples, we introduce a learnable sample synthesizer that optimizes the contrastive sample generation process during model training. SRA-CL adopts a plug-and-play design, enabling seamless integration with existing sequential recommendation architectures. Extensive experiments on four public datasets demonstrate the effectiveness and model-agnostic nature of our approach. Our code is available at https://github.com/ziqiangcui/SRA-CL

## 1 Introduction

Sequential recommendation aims to model user preferences based on historical behavior sequences, a task of significant value for online platforms like YouTube and Amazon. However, accurate preference modeling faces a fundamental challenge: data sparsity, as most users have only limited interaction records and most items receive little attention. To address this issue, numerous self-supervised learning techniques [39, 46, 34] have been proposed, leveraging auxiliary tasks to improve data utilization efficiency. Among these, contrastive learning has emerged as a predominant approach due to its conceptual simplicity and proven effectiveness [34, 24, 3, 22, 47]. Typically, it constructs positive sample pairs from the data and maximizes their agreement in the representation space [2].

As illustrated in Figure 1, existing contrastive learning approaches for sequential recommendation can be broadly classified into two categories: (1) inter-user contrastive learning, which contrasts sequences

---

[*]Equal contribution.
[†]Corresponding authors.

39th Conference on Neural Information Processing Systems (NeurIPS 2025).

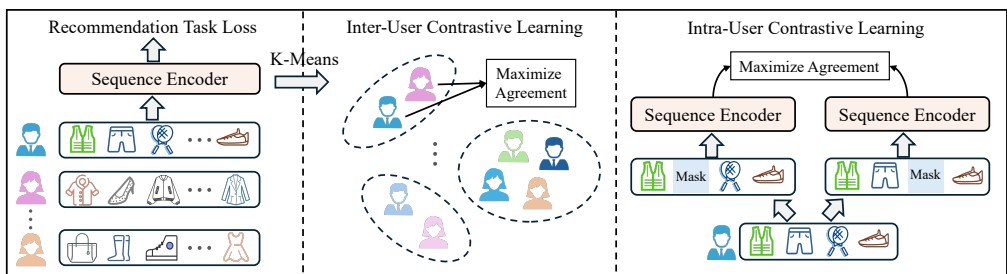

Figure 1: Illustration of existing contrastive learning methods in sequential recommendation, categorized into two main types: (1) inter-user contrastive learning and (2) intra-user contrastive learning.

from different users, and (2) intra-user contrastive learning, which contrasts different augmented views of a single user's sequence. In the inter-user paradigm, user sequence representations are clustered using K-means, and users within the same cluster are treated as positive samples for each other [3, 17, 23]. In the intra-user paradigm, perturbations are applied to a user's sequence to generate augmented views, and the similarity between these views is maximized [34, 21, 24, 22]. These contrastive learning methods are typically employed as auxiliary tasks alongside the primary recommendation objective and have been demonstrated to enhance recommendation performance by improving user representation learning [24, 36].

Despite their empirical success, existing methods suffer from several limitations in contrastive pair construction, which may undermine their effectiveness in recommendation scenarios. 1) **Semantic Divergence**. Many existing methods construct contrastive pairs through random augmentation operations such as random masking [34, 21] and Dropout [24]. However, in sequential recommendation where data is inherently sparse and exhibits sequential patterns, such random operations may lead to a complete change in the sequence's semantics (i.e., user preferences). Bringing semantically different sequences closer together in embedding space may diminish the model's ability to discriminate among distinct user preferences. Additionally, some methods determine contrastive pairs by clustering user representations derived from collaborative signals [3, 23], where users within the same cluster are considered positive pairs. However, the sparse ID signals can lead to low-quality representations and inaccurate clustering results. 2) **Unlearnability**. Existing methods rely on predefined rules to construct positive pairs, such as directly selecting users from the same cluster [3, 23, 21], or treating sequences sharing the same next item as positive pairs [24, 23]. These rigid heuristics impose strong assumptions that constrain models from autonomously learning optimal contrastive pairs. Moreover, the approach of using sequences with identical next items as positive pairs essentially replicates the recommendation objective (i.e., next-item prediction), providing no additional information gain. Therefore, the suboptimal construction of contrastive pairs in existing methods limits their effectiveness and hinders contrastive learning's full potential.

Given these limitations, constructing high-quality contrastive samples remains a critical challenge. Semantic information, which is readily available in textual data such as product categories, brands, and descriptions, provides a promising solution. Unlike sparse behavior signals, semantic data maintains validity regardless of data volume or training dynamics, as it derives from structured knowledge rather than co-occurrence patterns [46]. Additionally, semantic features offer complementary information beyond collaborative signals. Motivated by these advantages, we propose leveraging semantic information to construct superior contrastive pairs. However, accurately capturing user preferences requires models with powerful understanding and reasoning capabilities. Recent research has shown that large language models (LLMs) can effectively understand user preferences and achieve competitive performance on sequential recommendation tasks [35]. Inspired by this, we propose to enhance contrastive learning through LLM-powered semantic retrieval.

In this paper, we propose SRA-CL (Semantic Retrieval-Augmented Contrastive Learning), a novel framework with two key innovations: 1) **Semantic-based Retrieval**. We develop a semantic-based retrieval mechanism that operates at both inter-user and intra-user levels. For inter-user contrastive learning, we leverage LLMs to process sequential user interaction histories. Each sequence is fed to the LLM in chronological order of item interactions, where each item consists of both its attributes and textual description, enabling the model to generate preference-aware semantic embeddings through comprehensive understanding of user behavior patterns. For intra-user contrastive learning, we enhance item understanding by providing LLMs with both item attributes and their contextual

sequence information, producing context-aware semantic embeddings that capture both intrinsic item properties and their relevance within the recommendation context. Subsequently, we leverage the semantic embeddings to retrieve the top-$k$ most similar users and items, constructing candidate positive sample pools for contrastive learning. 2) **Learnable Sample Synthesis**. To construct more effective contrastive samples, our framework incorporates a learnable sample synthesizer. For inter-user contrastive learning, the synthesizer dynamically generates positive samples for each user sequence by selectively combining elements from the candidate pool. This generation process is jointly optimized with the model training, ensuring the synthesized samples effectively improve representation learning.

Our main contributions are summarized as follows.

- We propose a model-agnostic framework, SRA-CL, which leverages semantic information and the capabilities of LLMs to construct better contrastive pairs, thereby improving the contrastive learning in sequential recommendation.

- We propose a semantic-based retrieval approach for contrastive pair construction that integrates dual retrieval mechanisms: user retrieval for inter-user contrastive learning and item retrieval for intra-user contrastive learning, with each mechanism maintaining its dedicated candidate pool. To further enhance this framework, we introduce a learnable sample synthesizer that optimizes the contrastive sample generation process during model training.

- We conduct extensive experiments on four public datasets to validate the superiority and model-agnostic nature of our approach, as well as to confirm the efficacy of each module.

## 2 Preliminary

### 2.1 Sequential Recommendation Task

We denote the sets of users and items by $\mathcal{U}$ and $\mathcal{V}$, respectively. Each user $u \in \mathcal{U}$ has a chronological sequence of interacted items $\mathcal{S}_u = [v_1^u, v_2^u ..., v_n^u]$, where $v_t^u$ indicates the item that $u$ interacted with at step $t$, and $n$ is the predefined maximum sequence length. For user sequences longer than $n$, we retain only the most recent $n$ items. The goal of sequential recommendation is to predict the next item $v^+$ according to $\mathcal{S}_u$, which can be formulated as:

$$\arg\max_{v \in \mathcal{V}} P(v^+ = v | \mathcal{S}_u),\tag{1}$$

where the probability $P$ represents the likelihood of item $v$ being the next item, conditioned on $\mathcal{S}_u$.

### 2.2 Sequential Recommendation Backbone

Our method is model-agnostic and can be integrated with various sequential recommendation models, as demonstrated in Section 4.3. To facilitate the introduction of our approach, we adopt the transformer architecture [28] as the backbone recommendation model following previous studies [22–24].

**Embedding Layer.** We initialize an embedding matrix $\mathbf{M} \in \mathbb{R}^{|\mathcal{V}| \times d}$ to encode item IDs, where $|\mathcal{V}|$ represents the size of the item set and $d$ denotes the dimensionality of the latent space. Given a user interaction sequence $\mathcal{S}_u$, we obtain item embeddings $\mathbf{E}_u \in \mathbb{R}^{n \times d}$ and position embeddings $\mathbf{P} \in \mathbb{R}^{n \times d}$. Consequently, the input sequence $\mathcal{S}_u$ can be represented as $\mathbf{H}_u = \mathbf{E}_u + \mathbf{P}$.

**Sequence Encoder.** The representation of the input sequence is then fed into $L$ Transformer layers [28] to capture complex sequential patterns, which can be defined as follows:

$$\mathbf{H}_u^{(L)} = \text{Transformer}(\mathbf{H}_u), \quad \mathbf{h}_u = \mathbf{H}_u^{(L)}[-1].\tag{2}$$

Here, $\mathbf{h}_u \in \mathbb{R}^d$ represents the last position of $\mathbf{H}_u^{(L)}$ and is selected as the final representation of $\mathcal{S}_u$.

**Prediction and Objective Function.** During prediction, we calculate the probability of each item using $\hat{\mathbf{y}} = \text{softmax}(\mathbf{h}_u \mathbf{M}^{\text{T}})$, where $\hat{\mathbf{y}} \in \mathbb{R}^{|\mathcal{V}|}$ and $\hat{y}_v$ represents the likelihood of item $v$ being the next item. For training, we adopt the same cross-entropy loss function as our baseline methods [22–24] to ensure fairness, where $v^+$ denotes the ground truth item for user $u$.

$$\mathcal{L}_{\text{Rec}} = -\log(\hat{y}_{v^+}).\tag{3}$$

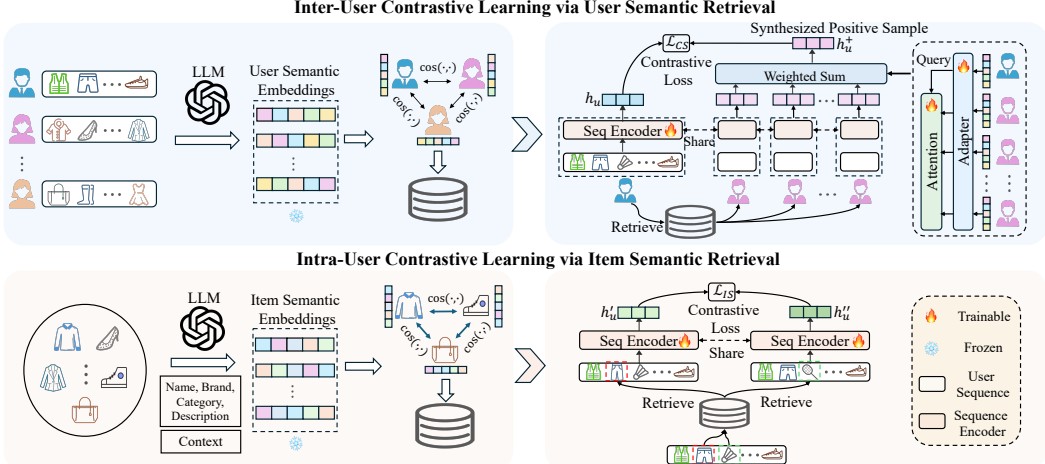

Figure 2: Overview of the proposed SRA-CL Framework.

# 3 The Framework of SRA-CL

In this section, we provide a detailed introduction to SRA-CL, which is shown in Figure 2. SRA-CL integrates inter-user contrastive learning via user semantic retrieval and intra-user contrastive learning via item semantic retrieval. To further enhance the framework, we introduce a learnable sample synthesizer that optimizes the contrastive sample generation process during model training.

## 3.1 Inter-User Contrastive Learning via User Semantic Retrieval

SRA-CL employs semantic retrieval to generate reliable supervision signals for inter-user contrastive learning. Leveraging the advanced reasoning capabilities of LLMs, we first derive a comprehensive representation of user preferences, which are then encoded as semantic embeddings. Based on the similarity of these embeddings, we introduce a semantic-based retrieval mechanism to construct a candidate sample pool. Subsequently, a learnable contrastive sample synthesis method is employed to generate effective contrastive pairs.

**User Preference Understanding with LLMs.** Textual data (e.g., product categories, brands, and descriptions) plays a pivotal role in recommender systems by encoding rich semantic signals that reflect user preferences. Given user $u$'s interaction sequence $\mathcal{S}_u$, we extract textual attributes for each item in chronological order, preserving both content and sequential context. These features are structured into a prompt $\mathcal{P}_u$, where item attributes and their order explicitly guide the LLM in inferring user preferences $\mathcal{A}_u = \text{LLM}(\mathcal{P}_u)$. The prompt template is detailed in Figure 6.

Next, we employ a pretrained text embedding model $\mathcal{M}$ to extract and convert the semantic information contained in the textual responses of LLMs into embeddings, which is formatted as:

$$\tilde{\mathbf{h}}_u = \mathcal{M}(\mathcal{A}_u), \tag{4}$$

where $\tilde{\mathbf{h}}_u \in \mathbb{R}^{\tilde{d}}$ represents the semantic embedding of user preferences and $\tilde{d}$ is the embedding size of the text embedding model $\mathcal{M}$. Specifically, $\mathcal{M}$ indicates SimCSE-RoBERTa [8] in this paper due to its open-source availability and excellent sentence semantic extraction capabilities. The generated semantic embeddings are cached and remain fixed throughout the whole training process.

**Semantic-based User Retrieval.** Once the semantic embeddings of user sequences are obtained, similar users can be retrieved based on semantic similarity. For a given user sequence $\mathcal{S}_u$, we calculate the cosine similarity between its semantic embedding $\tilde{\mathbf{h}}_u$ and the semantic embeddings of other users. Users are then ranked in descending order according to the computed semantic similarity. The top $k$ users are retrieved to construct the homogeneous user pool for user $u$, denoted as $\mathcal{N}_u$.

$$\mathcal{N}_u = \{u' \in \mathcal{U} \setminus \{u\} \mid \text{rank}(\text{cosine\_similarity}(\tilde{\mathbf{h}}_u, \tilde{\mathbf{h}}_{u'})) \leq k\}, \tag{5}$$

where $\mathcal{U} \setminus \{u\}$ denotes the set of all users except $u$.

**Learnable Contrastive Sample Synthesis.** Sole reliance on hard rules, such as selecting a user from the current user's dedicated candidate pool as the positive sample, often yields suboptimal solutions (as shown in Table 2). To enhance contrastive sample construction, we introduce a learnable sample synthesizer that optimizes the contrastive sample generation process during model training. Specifically, we first map the semantic representations of user sequences through a learnable adapter. Then, in the mapped space, we employ an attention mechanism, where the current user serves as a query to compute the probability $p_{u,u'}$ that each candidate user $u' \in \mathcal{N}_u$ is suitable as the positive sample for the current user $u$. This process is formulated as:

$$w_{u,u'} = \text{LeakyReLU}(\mathbf{a}^\top [\mathbf{W}\tilde{\mathbf{h}}_u \| \mathbf{W}\tilde{\mathbf{h}}_{u'}]), \tag{6}$$

$$p_{u,u'} = \text{softmax}_{u'}(w_{u,u'}) = \frac{\exp(w_{u,u'})}{\sum_{u_k \in \mathcal{N}_u} \exp(w_{u,u_k})}, \tag{7}$$

where $\mathbf{W} \in \mathbb{R}^{d \times \tilde{d}}$ is a learnable weight matrix, and $\|$ denotes the concatenation operation. $\mathbf{a} \in \mathbb{R}^{2d}$ represents a single-layer neural network used to generate the attention score, with the LeakyReLU activation function adopted [29]. The softmax function is employed to transform the coefficients into probabilities. Based on this, we generate the composite positive contrastive sample $\mathbf{h}_u^+$ for $\mathbf{h}_u$ by:

$$\mathbf{h}_u^+ = \sum\nolimits_{u' \in \mathcal{N}_u} p_{u,u'} \mathbf{h}_{u'}, \tag{8}$$

where $\mathbf{h}_{u'} \in \mathbb{R}^d$ is the recommendation model's output sequence representation for $u'$, as defined in Equation (2). This operation enables a fine-grained learnable selection of contrastive samples.

**Inter-User Contrastive Loss.** For each user $u$, $\mathbf{h}_u$ is the sequence representation obtained from the recommendation model. The synthetic representation $\mathbf{h}_u^+$ is regarded as the positive sample for $\mathbf{h}_u$, while the remaining $N - 1$ synthetic representations within the same batch are treated as negative samples for $\mathbf{h}_u$, where $N$ is the batch size. We compute the inter-user contrastive loss $\mathcal{L}_{\text{CS}}$ as follows:

$$\mathcal{L}_{\text{CS}} = -\log \frac{\exp\left(\mathbf{h}_u \cdot \mathbf{h}_u^+\right)}{\exp\left(\mathbf{h}_u \cdot \mathbf{h}_u^+\right) + \sum_{\mathbf{h}_u^- \in \mathbf{H}_u^-} \exp\left(\mathbf{h}_u \cdot \mathbf{h}_u^-\right)}, \tag{9}$$

where $(\cdot)$ represents the inner product operation, $\mathbf{H}_u^-$ denotes the set of negative samples for $\mathbf{h}_u$.

### 3.2 Intra-User Contrastive Learning via Item Semantic Retrieval

For intra-user contrastive learning, most existing methods apply predefined random perturbations to the original sequence to generate augmented views, which are treated as a pair of positive samples [34, 24]. A significant limitation of them is the introduction of considerable uncertainty in the semantic similarity between positive samples. This substantial variation in user sequence semantics among positive samples undermines the reliability of the contrastive learning process. To address this issue, we leverage a comprehensive understanding of both the semantic information of the item itself and the typical contexts in which the item appears. Based on this understanding, we replace certain items in the sequence with similar ones, resulting in semantic-consistent positive samples.

**Item Understanding with LLMs.** To enhance the LLM's comprehension of items, we provide two types of input information: (1) *textual attributes of the item*, including category, brand, and description, which supply fundamental information and enable the LLM to perform a coarse-grained assessment of item similarity; and (2) *user sequences containing the given item*. By analyzing the typical contexts in which an item appears, the LLM can infer the characteristics of its potential audience. This methodology facilitates more accurate evaluations of the relationships between items in the context of sequential recommendation. Given the token limit for LLM input, we have constrained the maximum number of item-related sequences in the prompt to 10, leaving the exploration of this value for future research. Next, these two types of information for item $v$ are integrated into a structured prompt $\mathcal{P}_v$, which is processed by the language model to generate the item summary $\mathcal{A}_v = \text{LLM}(\mathcal{P}_v)$. The detailed prompt template is illustrated in Figure 6. Then, the pretrained text embedding model $\mathcal{M}$ is used to convert the textual responses of LLMs into embeddings: $\tilde{\mathbf{e}}_v = \mathcal{M}(\mathcal{A}_v)$.

**Semantic-based Item Retrieval.** Similar to user retrieval, we compute the cosine similarity between the semantic embedding of an item and those of other items. Next, Top-$k$ most semantically relevant items for item $v$ are retrieved, which is formulated as:

$$\mathcal{N}_v = \{v' \in \mathcal{V} \setminus \{v\} \mid \text{rank}(\text{cosine\_similarity}(\tilde{\mathbf{e}}_v, \tilde{\mathbf{e}}_{v'})) \le k\}, \tag{10}$$

**Contrastive Sample Selection.** For intra-user contrastive learning, generating two semantic-consistent augmented views of the same user sequence is crucial. Here, we employ a semantic-based item substitution approach. Specifically, for each sequence $\mathcal{S}_u$, we randomly select 20% of the items. For each selected item $v$, we substitute it with a semantically similar item sampled from its candidate pool $\mathcal{N}_v$. This operation yields an augmented sequence $\mathcal{S}'_u$ derived from the original $\mathcal{S}_u$. By repeating this process, we obtain two augmented views, denoted as $\mathcal{S}'_u$ and $\mathcal{S}''_u$, which form a positive sample pair. Critically, the substitution is not entirely random but is guided by semantic similarity, which accounts for both item attribute similarity and contextual relevance in the recommendation scenario. This reduces uncertainty and enhances semantic consistency between augmented views.

Our preliminary experiments also explored the use of learnable synthesizers (analogous to inter-contrastive learning approaches) for generating substitute items, yet yielded no measurable performance improvements (shown in Table 4). This can be attributed to the inherently higher interpretability and quantifiability of item semantics relative to user preferences. Therefore, directly identifying appropriate substitutes from semantically similar candidate pools is simpler and more reliable compared to matching users with analogous preference patterns. A more detailed analysis is provided in Appendix C.1.

**Intra-User Contrastive Loss.** For the two augmented sequences $\mathcal{S}'_u$ and $\mathcal{S}''_u$, we obtain their hidden vectors $\mathbf{h}'_u$ and $\mathbf{h}''_u$ using the sequence encoder defined in Equation (2). Then the intra-user contrastive loss can be calculated as:

$$\mathcal{L}_{\text{IS}} = -\log \frac{\exp\left(\mathbf{h}'_u \cdot \mathbf{h}''_u\right)}{\exp\left(\mathbf{h}'_u \cdot \mathbf{h}''_u\right) + \sum_{\mathbf{h}^{\text{neg}}_u \in \mathbf{H}^{\text{neg}}_u} \exp\left(\mathbf{h}'_u \cdot \mathbf{h}^{\text{neg}}_u\right)} \tag{11}$$

In a batch with a size of $N$, we have $2N$ augmented sequences. Among these, $\mathbf{h}'_u$ and $\mathbf{h}''_u$ are positive samples of each other and are interchangeable. The remaining $2(N-1)$ samples excluding $\mathbf{h}'_u$ and $\mathbf{h}''_u$ are considered negative samples $\mathbf{H}^{\text{neg}}_u$.

---

**Algorithm 1** Training for SRA-CL

---

**Require:** Training data $\{\mathcal{S}_u\}$ for all $u \in \mathcal{U}$; hyperparameters $\alpha$, $\beta$, $k$
1: Obtain user semantic embeddings $\{\tilde{\mathbf{h}}_u\}$ for all $u \in \mathcal{U}$; obtain item semantic embeddings $\{\tilde{\mathbf{e}}_v\}$ for all $v \in \mathcal{V}$.
2: Freeze the embeddings $\{\tilde{\mathbf{h}}_u\}$ and $\{\tilde{\mathbf{e}}_v\}$, and initialize the model parameters.
3: **for** each iteration **do**
4:     Compute $\mathbf{h}_u$ using Equation (2).
5:     Calculate $\hat{\mathbf{y}} = \text{softmax}(\mathbf{h}_u \mathbf{M}^{\text{T}})$.
6:     Compute $\mathcal{L}_{\text{Rec}}$ using Equation (3).
7:     Retrieve $\mathcal{N}_u$ for each $u \in \mathcal{U}$ using Equation (5).
8:     Synthesize $\mathbf{h}^+_u$ for each $u$ using Equations (6), (7), and (8).
9:     Compute $\mathcal{L}_{\text{CS}}$ using Equation (9).
10:     Retrieve $\mathcal{N}_v$ for each $v \in \mathcal{V}$ using Equation (10).
11:     Generate $\mathcal{S}'_u$ and $\mathcal{S}''_u$, along with the corresponding $\mathbf{h}'_u$ and $\mathbf{h}''_u$.
12:     Compute $\mathcal{L}_{\text{IS}}$ using Equation (11).
13:     Calculate the total loss $\mathcal{L} = \mathcal{L}_{\text{Rec}} + \alpha\mathcal{L}_{\text{CS}} + \beta\mathcal{L}_{\text{IS}}$.
14:     Update the model parameters using the gradient of $\mathcal{L}$.
15: **end for**
16: Return the final model parameters $\theta$.

---

**Algorithm 2** Inference for SRA-CL

---

**Require:** Trained model parameters $\theta$; test data $\{\mathcal{S}_u\}$
1: **for** each user sequence in test data **do**
2:     Compute $\mathbf{h}_u$ using Equation (2).
3:     Calculate the predicted scores $\hat{\mathbf{y}} = \text{softmax}(\mathbf{h}_u \mathbf{M}^{\text{T}})$.
4:     Obtain the top-$k$ items with the highest scores in $\hat{\mathbf{y}}$.
5: **end for**
6: Return the recommended items for all users.

---

### 3.3 Training and Inference

During the training phase, all semantic embeddings are fixed. The training objective consists of three components: the loss of the recommendation model $\mathcal{L}_{\text{Rec}}$, which serves as the main loss, and the inter-user contrastive loss $\mathcal{L}_{\text{CS}}$ and intra-user contrastive loss $\mathcal{L}_{\text{IS}}$, which act as regularization terms.

$$\mathcal{L} = \mathcal{L}_{\text{Rec}} + \alpha\mathcal{L}_{\text{CS}} + \beta\mathcal{L}_{\text{IS}}, \tag{12}$$

where $\alpha$ and $\beta$ are hyperparameters.

During inference, only the recommendation backbone is utilized. The contrastive learning tasks and LLMs' semantic embeddings are not involved in the inference process. This implies that our framework can be deployed in real-world applications without incurring any additional inference latency from incorporating LLMs. The training and inference processes are detailed in Algorithm 1 and Algorithm 2, respectively.

## 4 Experiments

### 4.1 Experimental Settings

**Datasets.** Following previous studies [21, 34, 22], we conducted experiments on four public real-world datasets: Yelp, Amazon Sports, Beauty, and Office. The statistics for these datasets are presented in Table 3. More details about the datasets are shown in Appendix B.1.

**Evaluation Metrics.** To evaluate the performance of the models, we use widely recognized evaluation metrics: Hit Rate (HR) and Normalized Discounted Cumulative Gain (NDCG), follow previous studies [41, 30, 9, 12]. The leave-one-out strategy is employed, where the last interaction is used for testing, the second-to-last interaction for validation, and the remaining interactions for training. To ensure an unbiased evaluation, we rank the prediction on the whole item set without sampling.

**Baseline Methods.** We compare our method with 13 baseline methods, categorized into three groups: 1) *classical methods* (GRU4Rec [10], SASRec [12], BERT4Rec [27]), 2) *contrastive learning-based methods* ($\text{S}^3$-$\text{Rec}$ [46], CL4SRec [34], CoSeRec [21], ICLRec [3], DuoRec [24], MCLRec [22], ICSRec [23]), and 3) *LLM-based methods* (LRD [35], RLMRec [25], LLM-ESR [20]).

**Implementation Details.** All experiments are conducted with a single 32G V100 GPU. The embedding size is set to 64. We adopt the batch size of 256 and employ the Adam optimizer with a learning rate of 0.001. The dropout rate is set to 0.5 across all datasets. Following previous studies [35], we set the maximum sequence length to 20. The early stopping is applied if the metrics on the validation set do not improve over 10 consecutive epochs. For LLM, we use DeepSeek-V3 by invoking its API. We set the LLM's temperature $\tau$ to 0 and top-$p$ to 0.001. For the text embedding model $\mathcal{M}$, we use the pre-trained RoBERTa from Hugging Face. Note that identical settings are adopted for our method and baselines that involve LLMs and text embeddings to ensure fairness. More implementation details can be found in Appendix B.3.

### 4.2 Comparison Results with Baselines

The comparison results are presented in Table 1. Each experiment was conducted five times, and the average results are reported. SRA-CL consistently outperforms all baseline methods across all datasets, achieving performance improvements of up to 11.82%. The improvements are also confirmed by a paired t-test with a significance level of 0.01. Contrastive learning-based methods generally surpass traditional methods (GRU4Rec, SASRec, BERT4Rec). Among the contrastive learning baselines, MCLRec and ICSRec demonstrate superior performance. However, both methods underperform compared to SRA-CL, as they fail to control the quality of contrastive samples. SRA-CL mitigates this issue by introducing semantic-based retrieval augmentation, thereby improving the quality of contrastive samples and enhancing the overall effectiveness of contrastive learning. Regarding LLM-enhanced baselines, they demonstrate superior results compared to classical methods. However, our proposed SRA-CL achieves significant improvements over these LLM approaches. Unlike existing LLM-based methods, SRA-CL is fundamentally different in motivation—it specifically addresses the limitations in contrastive learning through enhanced construction of positive sample pairs using semantic information.

Table 1: Performance comparison of different methods on four datasets. Bold font indicates the best performance, while underlined values represent the second-best. SRA-CL achieves state-of-the-art results among all methods, as confirmed by a paired t-test with a significance level of 0.01. Due to space constraints, additional metrics (HR@10 and NDCG@10) are provided in Appendix C.2.

| Model | Yelp | | Sports | | Beauty | | Office | |
|---|---|---|---|---|---|---|---|---|
| | HR@20 | NDCG@20 | HR@20 | NDCG@20 | HR@20 | NDCG@20 | HR@20 | NDCG@20 |
| GRU4Rec | 0.0639 | 0.0243 | 0.0325 | 0.0129 | 0.0488 | 0.0189 | 0.0956 | 0.0361 |
| SASRec | 0.0899 | 0.0390 | 0.0498 | 0.0216 | 0.0887 | 0.0382 | 0.1329 | 0.0482 |
| BERT4Rec | 0.0913 | 0.0394 | 0.0578 | 0.0241 | 0.0933 | 0.0399 | 0.1436 | 0.0520 |
| $S^3$-Rec | 0.0964 | 0.0443 | 0.0607 | 0.0262 | 0.0994 | 0.0414 | 0.1568 | 0.0571 |
| CL4SRec | 0.0923 | 0.0395 | 0.0562 | 0.0235 | 0.0980 | 0.0416 | 0.1297 | 0.0488 |
| CoSeRec | 0.0984 | 0.0404 | 0.0638 | 0.0293 | 0.1034 | 0.0487 | 0.1354 | 0.0516 |
| ICLRec | 0.0974 | 0.0432 | 0.0636 | 0.0284 | 0.1056 | 0.0482 | 0.1513 | 0.0559 |
| DuoRec | 0.1173 | 0.0493 | 0.0706 | 0.0302 | 0.1224 | 0.0535 | 0.1549 | 0.0653 |
| MCLRec | 0.1150 | 0.0486 | 0.0736 | 0.0318 | 0.1239 | 0.0536 | 0.1629 | 0.0684 |
| ICSRec | 0.1165 | 0.0495 | 0.0728 | 0.0304 | 0.1205 | 0.0528 | 0.1643 | 0.0690 |
| LRD | 0.1082 | 0.0455 | 0.0589 | 0.0257 | 0.0931 | 0.0402 | 0.1468 | 0.0577 |
| RLMRec | 0.1125 | 0.0478 | 0.0664 | 0.0298 | 0.1190 | 0.0521 | 0.1532 | 0.0613 |
| LLM-ESR | 0.1061 | 0.0451 | 0.0638 | 0.0277 | 0.1064 | 0.0515 | 0.1425 | 0.0602 |
| **SRA-CL** | **0.1282** | **0.0533** | **0.0823** | **0.0347** | **0.1314** | **0.0568** | **0.1702** | **0.0725** |
| Improvement | 9.29% | 7.68% | 11.82% | 9.12% | 6.05% | 5.97% | 3.59% | 5.07% |

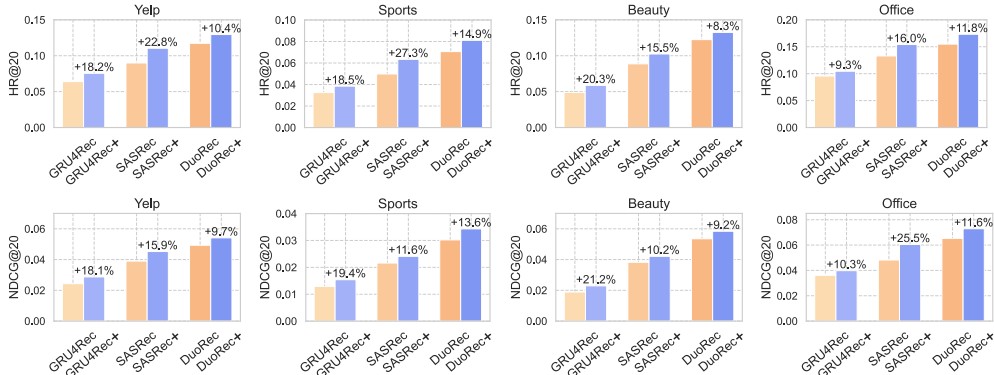

Figure 3: Experimental results demonstrating the model-agnostic nature and strong generalization capability of SRA-CL. "+" indicates the addition of SRA-CL to different recommendation models.

## 4.3 Validation of Model-Agnostic Characteristic

In this section, we validate the model-agnostic nature of our method. We select three classic recommendation models (GRU4Rec, SASRec, DuoRec) as the backbone and integrate SRA-CL to observe performance changes. We retain the original loss functions of the backbones and introduce our contrastive loss $\mathcal{L}_{CS}$ and $\mathcal{L}_{IS}$ during training. The results are shown in Figure 3, which indicate that for all three backbone methods, the versions enhanced with SRA-CL ("+") consistently outperform the original versions. Specifically, HR@20 improves by 8.3% to 27.3%, and NDCG@20 increases by 9.7% to 25.5%. These findings validate that SRA-CL can robustly improve the performance of various recommendation models.

## 4.4 Ablation Study

In this section, we evaluate the effectiveness of each component in SRA-CL. The results, presented in Table 2, demonstrate the impact of removing individual modules. Overall, the results show that removing any component degrades model performance, confirming the necessity of each module. Specifically, the variants "w/o $\mathcal{L}_{CS}$" and "w/o $\mathcal{L}_{IS}$" exhibit significant performance drops, highlighting the importance of both inter-user and intra-user contrastive learning objectives. The "w/o CL" variant suffers a more severe performance decline than those removing only one contrastive objective, suggesting that these two types of objectives complement each other. Additionally, the "w/o learn." variant also leads to reduced performance, indicating that a learning-based sample synthesizer is more effective than random selection for inter-user contrastive learning. Furthermore, removing

Table 2: Ablation study on all datasets.

| | Metric | w/o CL | w/o $\mathcal{L}_{CS}$ | w/o $\mathcal{L}_{IS}$ | w/o learn. | w/o semantic | w/o LLM | **Ours** |
|---|---|---|---|---|---|---|---|---|
| Yelp | H@20 | 0.1101 | 0.1203 | 0.1228 | 0.1253 | 0.1187 | 0.1190 | **0.1282** |
| | N@20 | 0.0473 | 0.0504 | 0.0519 | 0.0520 | 0.0495 | 0.0501 | **0.0533** |
| Sports | H@20 | 0.0745 | 0.0780 | 0.0795 | 0.0792 | 0.0772 | 0.0781 | **0.0823** |
| | N@20 | 0.0296 | 0.0315 | 0.0332 | 0.0336 | 0.0311 | 0.0314 | **0.0347** |
| Beauty | H@20 | 0.1206 | 0.1273 | 0.1279 | 0.1273 | 0.1265 | 0.1259 | **0.1314** |
| | N@20 | 0.0518 | 0.0546 | 0.0545 | 0.0551 | 0.0532 | 0.0537 | **0.0568** |
| Office | H@20 | 0.1476 | 0.1621 | 0.1619 | 0.1617 | 0.1624 | 0.1643 | **0.1702** |
| | N@20 | 0.0599 | 0.0691 | 0.0689 | 0.0681 | 0.0673 | 0.0692 | **0.0725** |

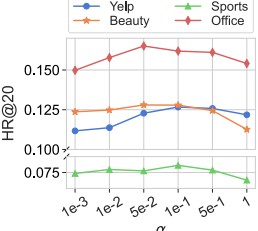 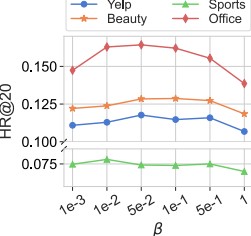 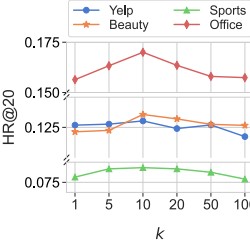

Figure 4: Hyperparameter experiments on the weight of $\mathcal{L}_{CS}$ ($\alpha$), the weight of $\mathcal{L}_{IS}$ ($\beta$), and the number of retrieved users/items ($k$).

semantic information and relying solely on collaborative signals for retrieval ("w/o semantic") results in a notable performance decline, underscoring the importance of semantic information in constructing high-quality contrastive samples. This finding aligns with our initial motivation. Similarly, the absence of LLM-based text processing ("w/o LLM") also results in performance degradation, demonstrating that utilizing the LLM's ability to understand and reason about user preferences is crucial.

### 4.5 Hyperparameter Study

In this section, we investigate the impacts of three key hyperparameters, $\alpha$, $\beta$, and $k$. Here, $\alpha$ and $\beta$ are the weights of $\mathcal{L}_{CS}$ and $\mathcal{L}_{IS}$, respectively, while $k$ denotes the number of retrieved users/items. From Figure 4, we observe that as both $\alpha$ and $\beta$ increase, the model's performance initially improves slightly and then decreases marginally. Empirically, the optimal range for $\alpha$ and $\beta$ is between 0.05 and 0.1. This is reasonable as contrastive learning loss acts as a regularization term. As the value of $k$ increases, the performance initially improves and then declines, with the optimal value around 10. As $k$ increases, the semantic relevance of retrieved neighbors decreases and randomness increases. A very small $k$ results in a candidate set that is too small without diversity. Conversely, a very large $k$ loses semantic relevance, thereby degrading the effectiveness of contrastive learning. Note that NDCG@20 results are provided in Figure 7 due to space limitation.

### 4.6 Contrastive Learning in Sparse Data: Analyzing SRA-CL's Superiority

To further examine SRA-CL's capability in mitigating the issue of low-quality contrastive samples in data-sparse scenarios, we categorize user sequences into three groups based on their length and compare the evaluation results of different methods. Due to space limitation, we present the experimental results for Beauty and Office, as shown in Figure 5. By comparing SRA-CL with the two strongest contrastive learning baselines (MCLRec and IC-SRec), we observe that SRA-CL consistently outperforms them across all user groups. No-

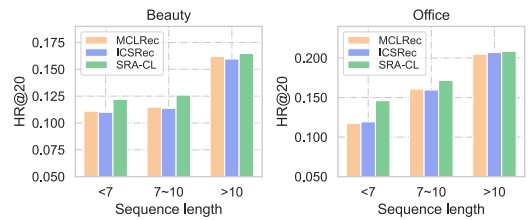

Figure 5: Performance comparison on different user groups among MCLRec, ICSRec and Ours.

tably, our method achieves greater improvements in sparser user groups (e.g., those with fewer than 7 or 7-10 interactions). This result further validates our motivation: while MCLRec and ICSRec

construct contrastive sample pairs based on collaborative signals, their performance degrades in data-sparse scenarios due to the diminished quality of contrastive samples. In contrast, our method significantly enhances the quality of contrastive pairs by incorporating semantic information, leading to superior performance under sparse data conditions.

## 5 Related Work

**Contrastive Learning in Sequential Recommendation.** Contrastive learning has been successfully used to enhance sequential recommendation [40, 11, 38, 31, 33, 37, 4, 5, 23]. In terms of the composition of contrastive samples, we categorize existing methods into two types: (1) Inter-user. This involves generating contrastive samples from different user sequences. For example, ICLRec [3] clusters user interests into distinct categories by K-Means and brings the representations of users with similar interests closer. ICSRec [23] further segments a user's behavior sequence into multiple subsequences to generate finer-grained user intentions for contrastive learning. These methods generate contrastive supervision signals based on collaborative signals. However, the sparsity of the co-occurrence pattern leads to unreliable clustering results, which in turn affects the performance of contrastive learning. (2) Intra-user. This involves applying perturbations to the original sequence to generate augmented views. The two views of the same sequence are treated as a pair of positive samples. For example, CL4SRec [34] employs three data-level augmentation operators: Cropping, Masking, and Reordering, to create contrastive pairs. CoSeRec [21] introduces two additional informative augmentation operators, building upon the foundation of CL4SRec. In addition, some methods generate augmented views from the model's hidden layers. A notable example is DuoRec [24], which creates positive pairs by forward-passing a sequence representation twice with different dropout masks. MCLRec [22] further combines data-level and model-level augmentation. Despite their effectiveness, they employ random operators, introducing significant uncertainty and potentially generating unreasonable positive samples for contrastive learning.

**Sequential Recommendation with LLMs.** Building upon foundations laid by traditional recommender systems [9, 15, 16], recent studies have successfully integrated LLMs into the recommender paradigm [32, 43, 14, 6]. Overall, LLMs are employed either as direct recommenders or as tools for extracting semantic information [13, 19, 42, 1, 26]. In the former approach, all inputs are converted into textual format, and the LLM generates recommendations based on its pre-trained knowledge or after undergoing supervised fine-tuning. Representative examples include LC-Rec [44], LLM-TRSR [45], and CALRec [18]. However, these methods rely on the inference process of large language models to generate recommendation results, which is computationally expensive and often challenging to deploy in practical scenarios. Another line of research [26, 30, 20, 35, 42, 1] leverages LLMs to process semantic information and incorporates it into traditional ID-based models. For example, SLIM [30] distills knowledge from large-scale LLMs into a smaller student LLM to improve the recommendation model. LLM-ESR [20] addresses the long-tail problem by leveraging collaborative signals and semantic information through dual-view modeling and self-distillation. LRD [35] utilizes the LLM to explore potential relations between items and reconstructs one item based on its relation to another. Unlike the aforementioned methods, our approach, grounded in the essence of contrastive learning, aims to construct more effective contrastive pairs with LLMs.

## 6 Conclusion

In this paper, we analyze the limitations of contrastive learning in sequential recommendation, namely Semantic Divergence and Unlearnability. To address these issues, we propose SRA-CL, a novel framework that enhances contrastive sample construction by integrating LLM-based semantic retrieval with a learnable sample synthesizer. SRA-CL leverages the capabilities of LLMs without increasing the inference time of the recommendation model, making it practical for large-scale real-world applications. Through comprehensive experiments, we demonstrate that LLM-based semantic-guided contrastive sample construction improves the contrastive learning, and we validate the effectiveness of the learnable sample synthesis mechanism. Furthermore, experiments with different recommendation model backbones confirm the model-agnostic nature of our approach.

## 7 Acknowledgements

This work was supported by the Early Career Scheme (No. CityU 21219323) and the General Research Fund (No. CityU 11220324) of the University Grants Committee (UGC), the NSFC Young Scientists Fund (No. 9240127), and the Donation for Research Projects (No. 9220187 and No. 9229164).

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

# A  Technical Supplement to SRA-CL

## A.1  Prompt Template

In this section, we provide a detailed description of the LLM prompt templates employed in our study. Specifically, to enhance the model's ability to comprehend user preferences and items, we have meticulously designed specialized prompts, as illustrated in Figure 6.

---

**Prompt Template**

**User**

You are a recommender system expert. A user's historical interactions are provided below in chronological order:
<product_name_1, brand_1, category_1, description_1>;
<product_name_2, brand_2, category_2, description_2>;
...
<product_name_n, brand_n, category_n, description_n>

Please analyze both the textual information and sequential patterns, and summarize the user preferences with no more than 200 words.

**Item**

Assume you are a recommender system expert. Below is the information of a specific product.
The product name is <name>; category is <cate>; brand is <brand>; description is < description>.
The historical sequences of users who have interacted with this product:
{sequence1};
{sequence2};
…
{sequence10}.

Please describe the given item and its potential audience according to its attributes and context. Your response should not exceed 100 words.

---

Figure 6: Prompt Template.

## A.2  Efficiency Analysis

**Inference Efficiency**. During inference, only the recommendation backbone is utilized. The contrastive learning tasks and the LLMs' semantic embeddings are not involved in the inference process. This ensures that our framework can be deployed in real-world applications without incurring any additional inference latency from incorporating LLMs.

**Training Efficiency**. The training process of our method consists of two stages: In the first stage, we use an LLM API to obtain semantic information and convert it into embeddings, which are then cached to construct contrastive sample indices. The primary time cost in this stage comes from the API calls. However, by employing asynchronous concurrency, this step can be completed within a few hours. Crucially, this stage is performed once and requires no repetition during model training. In the second stage, we use the pre-constructed contrastive sample index to train the recommendation model. Regarding the computational complexity of this stage, our method maintains comparable time complexity to general ID-based contrastive recommendation approaches. The only additional overhead during training compared to conventional contrastive recommendation models comes from the lightweight learnable sample synthesis module whose parameter size is negligible compared to that of the main recommendation model.

# B  Experimental Setting Details

## B.1  Datasets

We conducted experiments on four public real-world datasets: Yelp, Sports, Beauty, and Office. The statistics for these datasets are presented in Table 3. These datasets cover a diverse range of

Table 3: Dataset statistics.

| Datasets | #Users | #Items | #Actions | Avg. Length | Density |
|---|---|---|---|---|---|
| Yelp | 19,936 | 14,587 | 207,952 | 10.4 | 0.07% |
| Sports | 35,598 | 18,357 | 296,337 | 8.3 | 0.05% |
| Beauty | 22,363 | 12,101 | 198,502 | 8.8 | 0.07% |
| Office | 4,905 | 2,420 | 53,258 | 10.9 | 0.45% |

application scenarios. The Yelp dataset[3] is widely used for business recommendations. The Sports, Beauty, and Office datasets are sourced from Amazon[4], one of the largest e-commerce platforms. Following previous studies [21, 34, 22], the users and items that have fewer than five interactions are removed.

## B.2 Baseline Methods

To ensure a comprehensive assessment, we compare our method with 12 baseline methods, categorized into three groups: classical methods (GRU4Rec, SASRec, BERT4Rec), contrastive learning-based methods ($S^3$-Rec, CL4SRec, CoSeRec, ICLRec, DuoRec, MCLRec, ICSRec), and LLM-enhanced methods (LRD, LLM-ESR).

- **GRU4Rec** [10] applies recurrent neural networks (RNN) to sequential recommendation.
- **SASRec** [12] is the first work to utilize the self-attention mechanism for sequential recommendation.
- **BERT4Rec** [27] employs the BERT [7] framework to capture the context information of user behaviors.
- **$S^3$-Rec** [46] leverages four self-supervised objectives to uncover the inherent correlations within the data.
- **CL4SRec** [34] proposes three random augmentation operators to generate positive samples for contrastive learning.
- **CoSeRec** [21] introduces two additional informative augmentation operators, building upon the foundation of CL4SRec.
- **ICLRec** [3] clusters user interests into distinct categories and brings the representations of users with similar interests closer together.
- **DuoRec** [24] combines a model-level dropout augmentation and a sampling strategy for choosing hard positive samples.
- **MCLRec** [22] integrates CL4SRec's random data augmentation for the input sequence and employs MLP layers for model-level augmentation.
- **ICSRec** [23] is an improvement on ICLRec, further segmenting a user's sequential behaviors into multiple subsequences to generate finer-grained user intentions for contrastive learning.
- **LRD** [35] is an LLM-based method. It leverages LLMs to discover new relations between items and reconstructs one item based on the relation and another item.
- **RLMRec** [25] utilizes LLMs to generate text profiles and combine their semantic embeddings with recommendation models.
- **LLM-ESR** [20] is also an LLM-based method. It addresses the long-tail problem by simultaneously leveraging collaborative signals and semantic information through the dual-view modeling and self-distillation.

## B.3 Implementation Details

All experiments are conducted with a single V100 GPU. The embedding size for all methods is set to 64 for a fair comparison. We use a training batch size of 256 and employ the Adam optimizer

---

[3]https://www.yelp.com/dataset
[4]http://jmcauley.ucsd.edu/data/amazon/

with a learning rate of 0.001. The dropout rate is set to 0.5 for both the embedding layer and the hidden layers across all datasets. Following previous studies [35], we set the maximum sequence length to 20 for all datasets. The early stopping is applied if the metrics on the validation set do not improve over 10 consecutive epochs. Our method is model-agnostic and can be applied to any sequential recommendation model. The transformer backbone mentioned in Sec. 2.2 comprises two layers, each with two attention heads. For the LLM, we select DeepSeek-V3, a robust large language model that demonstrates exceptional performance on both standard benchmarks and open-ended generation evaluations. For detailed information about DeepSeek, please refer to their official website[5]. Specifically, we utilize DeepSeek-V3 by invoking its API[6]. To reduce text randomness of the LLM, we set the temperature $\tau$ to 0 and the top-$p$ to 0.001. For the text embedding model $\mathcal{M}$, we use the pre-trained SimCSE-RoBERTa[7] from Hugging Face. Identical settings are used for baselines that involve LLMs and text embeddings to ensure fairness.

# C  Additional Results & Analysis

## C.1  Discussion on Learnable Sample Synthesis

**Inter-User Contrastive Learning**. User preferences exhibit significant heterogeneity across individuals. Sole reliance on hard rules, such as selecting a user from the current user's dedicated candidate pool as the positive sample, may yield suboptimal solutions. Our experiments (as shown in Table 2 "w/o learn.") validated this. To enhance contrastive sample construction, we introduce a learnable sample synthesizer that optimizes the contrastive sample generation process during model training for inter-user contrastive learning.

**Intra-User Contrastive Learning**. Our preliminary experiments also explored the use of learnable synthesizers (analogous to inter-contrastive learning approaches) for generating substitute items, yet yielded no measurable performance improvements (shown in Table 4). Our analysis suggests this results from the inherent nature of item semantics being more readily interpretable and quantifiable than user preferences. Therefore, directly identifying appropriate substitutes from semantically similar candidate pools is simpler and more reliable compared to matching users with analogous preference patterns.

Table 4: Performance impact of learnable versus non-learnable sample synthesis strategies in **intra-user** contrastive learning.

|  | Yelp | | Sports | | Beauty | | Office | |
|---|---|---|---|---|---|---|---|---|
|  | HR@20 | NDCG@20 | HR@20 | NDCG@20 | HR@20 | NDCG@20 | HR@20 | NDCG@20 |
| Learnable | 0.1276 | 0.0531 | 0.0825 | 0.0344 | 0.1309 | 0.0561 | 0.1706 | 0.0722 |
| Unlearnable | 0.1282 | 0.0533 | 0.0823 | 0.0347 | 0.1314 | 0.0568 | 0.1702 | 0.0725 |

## C.2  Additional Comparison Results

We provide additional comparison results (HR@10 and NDCG@10) of different methods in Table 5. The experimental results demonstrate that our method outperforms all baselines across all datasets, further validating its superiority.

## C.3  Additional Results for Hyperparameter Experiments

Due to space constraints, we only present HR@20 in Figure 4 of the main text for hyperparameter study. Here, we additionally report the NDCG@20 evaluation results in Figure 7, providing complementary performance metrics for comprehensive analysis. As shown, the trend in NDCG@20 closely aligns with that of HR@20.

---

[5]https://github.com/deepseek-ai/DeepSeek-V3
[6]https://api-docs.deepseek.com/
[7]https://huggingface.co/princeton-nlp/sup-simcse-roberta-large

Table 5: Additional comparison results for HR@10 and NDCG@10. Bold font indicates the best performance, while underlined values represent the second-best. "ND" represents for "NDCG". Our method SRA-CL achieves state-of-the-art results among all methods, as confirmed by a paired t-test with a significance level of 0.01.

| Model | Yelp | | Sports | | Beauty | | Office | |
|---|---|---|---|---|---|---|---|---|
| | HR@10 | ND@10 | HR@10 | ND@10 | HR@10 | ND@10 | HR@10 | ND@10 |
| GRU4Rec | 0.0362 | 0.0173 | 0.0193 | 0.0096 | 0.0279 | 0.0137 | 0.0540 | 0.0260 |
| SASRec | 0.0572 | 0.0308 | 0.0304 | 0.0157 | 0.0612 | 0.0336 | 0.0791 | 0.0348 |
| BERT4Rec | 0.0582 | 0.0311 | 0.0349 | 0.0189 | 0.0628 | 0.0352 | 0.0821 | 0.0376 |
| $S^3$-Rec | 0.0612 | 0.0339 | 0.0385 | 0.0204 | 0.0647 | 0.0327 | 0.0931 | 0.0426 |
| CL4SRec | 0.0583 | 0.0315 | 0.0358 | 0.0189 | 0.0649 | 0.0329 | 0.0695 | 0.0322 |
| CoSeRec | 0.0607 | 0.0309 | 0.0439 | 0.0244 | 0.0725 | 0.0410 | 0.0782 | 0.0412 |
| ICLRec | 0.0598 | 0.0328 | 0.0428 | 0.0235 | 0.0713 | 0.0396 | 0.0922 | 0.0411 |
| DuoRec | 0.0747 | 0.0380 | 0.0474 | 0.0242 | 0.0841 | 0.0443 | 0.1015 | 0.0519 |
| MCLRec | 0.0721 | 0.0378 | 0.0498 | 0.0257 | 0.0870 | 0.0442 | 0.1036 | 0.0538 |
| ICSRec | 0.0738 | 0.0380 | 0.0487 | 0.0243 | 0.0844 | 0.0437 | 0.1034 | 0.0540 |
| LRD | 0.0693 | 0.0357 | 0.0376 | 0.0191 | 0.0620 | 0.0294 | 0.0887 | 0.0431 |
| RLMRec | 0.0709 | 0.0371 | 0.0426 | 0.0238 | 0.0764 | 0.0439 | 0.0927 | 0.0496 |
| LLM-ESR | 0.0669 | 0.0353 | 0.0415 | 0.0221 | 0.0750 | 0.0435 | 0.0889 | 0.0468 |
| **SRA-CL** | **0.0817** | **0.0419** | **0.0539** | **0.0274** | **0.0924** | **0.0469** | **0.1111** | **0.0575** |
| Improvement | 9.37% | 10.26% | 8.23% | 6.61% | 6.21% | 6.11% | 7.24% | 6.48% |

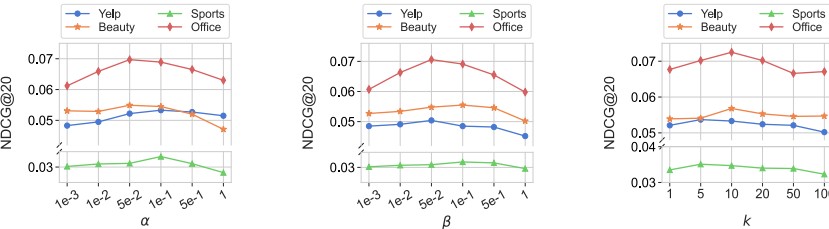

Figure 7: Hyperparameter experiments on the weight of $\mathcal{L}_{CS}$ ($\alpha$), the weight of $\mathcal{L}_{IS}$ ($\beta$), and the number of retrieved users/items ($k$) (NDCG results).

# D  Other Discussions

## D.1  Limitation

Considering computational budgets and resource limitations, we specifically analyzed how two selected LLMs (DeepSeek and Qwen) affect our framework's effectiveness. While more LLMs might yield different results, our study focused on these representative models.

## D.2  Broader Impacts

SRA-CL demonstrates significant improvements in sequential recommendation accuracy (positive impact), with potential applicability to real-world platforms. Like all recommendation systems, its personalized nature may occasionally limit content diversity, though this effect is inherent to the recommendation paradigm rather than unique to our method.

