# OpenReview forum: "Semantic Retrieval Augmented Contrastive Learning for Sequential Recommendation"
_NeurIPS.cc/2025/Conference — NeurIPS 2025 poster_

### Official Review · Reviewer_UspK · 2025-07-02

**Clarity:** 2
**Significance:** 2
**Originality:** 2
**Rating:** 4
**Confidence:** 4

**Summary:**

This paper leverages the semantic understanding capabilities of LLMs to enhance contrastive learning in sequential recommendation systems. It proposes a semantic similarity-based contrastive pair construction framework called SRA-CL. The core idea is to utilize LLM-generated semantic embeddings to construct positive contrastive pairs through semantic-based retrieval, followed by a learnable sample synthesizer that optimizes the contrastive sample generation process during model training.

The motivation behind using semantic information instead of random perturbations or sparse collaborative signals is to address the semantic divergence problem and the learnable sample synthesis component aims to overcome the unlearnability issue by enabling models to autonomously learn optimal contrastive pairs rather than relying on predefined heuristic rules. SRA-CL adopts a plug-and-play design that can seamlessly integrate with existing sequential recommendation architectures, demonstrating both effectiveness and model-agnostic nature across multiple datasets.

**Questions:**

Q1: Experimental Validation of the Learnable Sample Synthesizer's Effectiveness

Authors propose a Learnable Contrastive Sample module to address user preference heterogeneity, but the experiments lack direct evidence that current positive pair construction methods suffer from this problem. Please provide:
- Visualization of selection probabilities for positive samples generated by the learnable synthesizer
- Comparative analysis showing how the synthesizer's selections differ from baseline methods

Q2:Comparison with State-of-the-Art LLM-Based Embedding Methods.



if authors could partially solve my concerns during the rebuttal period, I am pleased to raise my score.

**Ethical Concerns:**

["NO or VERY MINOR ethics concerns only"]

**Final Justification:**

I appreciate the authors' efforts during the rebuttal period. The newly added experiments better support the paper’s motivation, and the inclusion of results with different embedding models further demonstrates the effectiveness of SRA-CL. These clarifications have resolved most of my earlier concerns. Accordingly, I am raising my score to borderline accept.

**Limitations:**

yes

**Quality:**

3

**Strengths And Weaknesses:**

S1: Problem Scope
This paper aims to the construction of high-quality contrastive pairs for self-supervised learning by leveraging LLM semantic understanding.

S2: Experimental Validation
The authors conduct experiments across four public datasets to validate both the effectiveness and model-agnostic nature of the approach. The experimental design includes ablation studies to confirm the efficacy of individual components, providing evidence for the contribution of each proposed module to the overall performance improvement.

W1: Insufficient Justification of Motivation.

The authors claim that existing methods for constructing contrastive pairs may lead to a complete change in sequence semantics (i.e., user preferences). However, the approach of selecting positive pairs based on LLM preference summarization and semantic embedding retrieval still exhibits a significant gap between LLM-generated preferences and actual user preferences [1,2]. The proposed method does not adequately address this fundamental motivation. This limitation is further evidenced in Figure 4, where both small and large numbers of retrieved semantically-related users/items as contrastive learning samples result in suboptimal experimental results.

[1] Ren, Xubin, et al. "Representation learning with large language models for recommendation." Proceedings of the ACM Web Conference 2024. 2024.
[2] Sheng, Leheng, et al. "Language Representations Can be What Recommenders Need: Findings and Potentials." arXiv preprint arXiv:2407.05441 (2024).


W2: Lack of Experimental Validation for Proposed Method's Motivation.

The authors propose a Learnable Contrastive Sample module to address the issue that "User preferences exhibit significant heterogeneity across individuals." However, the experiments fail to demonstrate that current positive pair construction methods actually suffer from this problem. The authors should provide visualization of the selection probabilities for positive samples generated by this module to further substantiate their claims.


W3: Experimental Design Fails to Incorporate Recent Advances
The paper's embedding retrieval still relies on early small-scale embedding models. Recent research has demonstrated that LLM-based embeddings, such as LLaMA [4], can significantly enhance model performance, as evidenced in LLM-ESR [3] and AlphaRec [2]. The authors are recommended to supplement their work with relevant experiments using state-of-the-art embedding approaches.

[3] Liu, Qidong, et al. "Llm-esr: Large language models enhancement for long-tailed sequential recommendation." Advances in Neural Information Processing Systems 37 (2024): 26701-26727.
[4] Touvron, Hugo, et al. "Llama: Open and efficient foundation language models." arXiv preprint arXiv:2302.13971 (2023).

---

> ### Author Rebuttal · Authors · 2025-07-31
>
> We sincerely appreciate the reviewer's valuable time and insightful comments. Below are our point-by-point responses to the questions raised. We look forward to any additional feedback the reviewer may provide.
>
>
> # Response to W1: Justification of Motivation
> We sincerely apologize for any misunderstanding, and we appreciate this opportunity to clarify our motivation.
> - As presented in Page 2, lines 41–44, we posit: "*Many existing methods construct contrastive pairs through random augmentation operations such as random masking and Dropout. However, in sequential recommendation where data is inherently sparse and exhibits sequential patterns, such random operations may lead to a complete change in the sequence’s semantics*." Specifically, our critique focuses on random data augmentation methods, wherein contrastive samples are constructed in an entirely arbitrary fashion, frequently compromising the semantic of the original sequence. In contrast, our method is based on LLM's semantic comprehension. It stands to reason that our approach would yield more reliable results. This claim has been empirically validated through both ablation studies (Table 2  "w/o semantic") and comparative experiments (Table 1).
>
> - To further mitigate the potential gap between LLM-summarized user preferences and genuine user interests, we avoid naively adopting LLM-identified "most relevant" users as positive contrastive samples. Instead, we propose a Learnable Sample Synthesis approach, where the LLM only performs initial screening to generate a candidate set of potential positive samples, and then a trainable synthesis mechanism is used to dynamically generate informative samples from the candidate set, which is optimized end-to-end based on the final recommendation performance. This ensures the synthesized positive samples enhance recommendation metrics.
>
> - Explanation of **Figure 4**. The figure illustrates the impact of k on model performance. When k=1, the method simply adopts the top-1 most relevant sample selected by the LLM as the contrastive sample, bypassing our learnable synthesis module. The suboptimal performance at this setting indicates a potential gap between the LLM's inferred user preferences and actual user preferences, underscoring that leveraging LLMs to guide contrastive sample construction is indeed nontrivial. As k increases (e.g., to k=10), the LLM performs preliminary screening to select k potentially relevant candidates for the sample synthesis module. Through end-to-end learning optimization, our module synthesizes an optimal contrastive sample from these candidates. The performance improvement indicates that the learnable synthesizer effectively mitigates the semantic gap between LLM outputs and real user preferences, thereby generating superior contrastive samples to improve the performance. Excessively large k values (k=100) relax the LLM's semantic similarity threshold, introducing substantial noise, which compromises the learning process and degrades performance. Overall, this figure highlights our key contribution: developing an effective learnable approach to bridge the semantic gap between LLM outputs and genuine user preferences.
>
>
> # Response to W2 & Q1:  Experimental Validation of the Learnable Sample Synthesizer
> ## 1) Clarification of Motivations
> - The statement, "User preferences exhibit significant heterogeneity across individuals," highlights the diversity of user interests. The primary goal of recommendation models is to effectively model this diverse preferences. However, existing methods construct contrastive pairs through random augmentation or predefined hard rules, which significantly alter the user preferences, potentially leading to substantial differences in preferences between two users treated as positive samples. Pulling users with differing interests closer in the latent space leads to representation homogenization, which contradicts the core objective of recommendation.
>
> - Our method leverages LLM-based semantic information to perform an initial screening and generate a candidate set that aligns with each user's preference-related semantics, which helps alleviate the aforementioned issue. However, directly using the top-1 sample as the positive sample for contrastive learning may lead to a potential problem: “There may exist a gap between the LLM’s interpretation of preferences and the user’s actual preferences.” To address this issue, we propose the Learnable Contrastive Sample Synthesis module.
>
> ## 2) Experimental Validation
> In the ablation study section of our original paper (Table 2), we reported the experimental results for the "w/o learn." setting. The observed performance degradation validates the effectiveness of the proposed Learnable Contrastive Sample Synthesis module.
>
> We greatly appreciate the reviewer’s suggestion. In response, we conducted two additional experiments to validate the motivation, which will be added to our final version.
>
> - To validate whether there is a gap between the LLM's understanding of user preferences and the actual user preferences, we randomly selected 100 users and, for each user, observed and compared the semantic similarity score distribution of the users in the candidate set (as derived from LLM-based relevance) with the weight distribution learned by the proposed learnable module for these users. The results show that the ranking results differ in 99% of the cases. Due to space limitations, we provide an example of the selection probabilities comparison as follows:
>
> |                                           | u1    | u2    | u3    | u4    | u5    | u6    | u7    | u8    | u9    | u10   |
> |-------------------------------------------|-------|-------|-------|-------|-------|-------|-------|-------|-------|-------|
> | similarity scores wihout learn.           | 0.882 | 0.856 | 0.854 | 0.819 | 0.798 | 0.785 | 0.763 | 0.744 | 0.740 | 0.719 |
> | attention scores with learn. (normalized) | 0.051 | 0.099 | 0.198 | 0.152 | 0.067 | 0.101 | 0.120 | 0.072 | 0.061 | 0.079 |
>
> From the results, we can observe that there are clear differences between the two distributions, which validates the rationale behind employing the proposed learnable module. Our learnable module leverages a non-linear adapter and an attention mechanism to automatically assign weights to the samples in the LLM-filtered candidate set, which makes that the selection probabilities of candidate samples are no longer solely based on the original semantic embeddings but are instead learned and optimized end-to-end according to the objective of the recommendation model.
>
> - To validate the impact of the learnable module on representation heterogenization, we visualized the user sequence representations learned by our method with and without the learnable module using t-SNE. Due to rebuttal guidelines prohibiting the inclusion of images or links, we provide a brief summary of the results here. The embeddings with the learnable module are more dispersed, indicating greater heterogeneity in the representations, while the embeddings without the module are relatively compact. The results suggest that the learnable module effectively mitigates the risk of user representation homogenization or even collapse, preserving the heterogeneous diversity of user interests.
>
>
> # Response to W3 & Q2: SOTA LLM-based Text Embedding Models
>
> We sincerely appreciate the reviewer's suggestions regarding the text embedding model. We selected sup-simcse-roberta-large for two reasons: 1) It is a widely adopted model for extracting sentence-level semantic embeddings, and 2) Since this component is not our primary innovation focus, using a general embedding model better demonstrates that our method's performance does not rely on advanced text embedding models.
>
> We fully agree that comparing LLM-based text embedding models would yield valuable insights. In response, we have extended our evaluation to incorporate two additional SOTA embedding models: 1) OpenAI’s **text-ada-embedding-002**, and 2) **Qwen3-Embedding-8B**. The results are shown as the following table:
>
> |                                        |  Yelp  |         | Sports |         | Beauty |         | Office |         |
> |----------------------------------------|:------:|:-------:|:------:|:-------:|:------:|:-------:|:------:|:-------:|
> |  | HR@20  | NDCG@20 | HR@20  | NDCG@20 | HR@20  | NDCG@20 | HR@20  | NDCG@20 |
> | simcse-roberta-large | 0.1282 | 0.0533  | 0.0823 | 0.0347  | 0.1314 | 0.0568  | 0.1702 | 0.0725  |
> | OpenAI/text-ada-embedding-002          | 0.1296 | 0.0540  | 0.0835 | 0.0354  | 0.1331 | 0.0576  | 0.1718 | 0.0730  |
> | Qwen/Qwen3-Embedding-8B                | 0.1327 | 0.0556  | 0.0858 | 0.0375  | 0.1362 | 0.0593  | 0.1746 | 0.0748  |
>
>
> Our results show that advanced text embedding models can boost the performance of our method: 1) text-ada-embedding-002 (1B params, 1536D) outperforms SimCSE-RoBERTa-large (335M params, 1024D) across all datasets, thanks to its greater capacity for semantic representation. 2) Qwen3-Embedding-8B (8B params, 4096D) achieves SOTA results, surpassing both models in preference modeling accuracy.
>
> Furthermore, to demonstrate that our method consistently outperforms other LLM-based approaches across different text embedding models, we compared our method with the strongest LLM baseline, RLMRec, under the best-performing Qwen3-Embedding-8B.
>
> |                                    |  Yelp  |         | Sports |         | Beauty |         | Office |         |
> |------------------------------------|:------:|:-------:|:------:|:-------:|:------:|:-------:|:------:|:-------:|
> |                                    | HR@20  | NDCG@20 | HR@20  | NDCG@20 | HR@20  | NDCG@20 | HR@20  | NDCG@20 |
> | RLMRec with Qwen3-Embedding-8B     | 0.1146 | 0.0492  | 0.0690 | 0.0319  | 0.1244 | 0.0545  | 0.1571 | 0.0652  |
> | Our method with Qwen3-Embedding-8B | 0.1327 | 0.0556  | 0.0858 | 0.0375  | 0.1362 | 0.0593  | 0.1746 | 0.0748  |

---

> ### Author Response · Authors · 2025-08-06
>
> Dear Reviewer UspK,
>
> We sincerely appreciate the time and effort you've dedicated to reviewing our work. We have carefully addressed all your comments in our rebuttal, including detailed explanations and experiments regarding:
>
> - The experimental validation of our motivation, and
>
> - The SOTA LLM-based embedding models.
>
> Please let us know if these responses have adequately addressed your concerns.
>
> Thank you again for your valuable insights. We look forward to your reply.

---

> > ### Comment · Reviewer_UspK · 2025-08-08
> >
> > I appreciate the authors' efforts during the rebuttal period. The newly added experiments better support the paper’s motivation, and the inclusion of results with different embedding models further demonstrates the effectiveness of SRA-CL. These clarifications have resolved most of my earlier concerns. Accordingly, I am raising my score to borderline accept.

---

> > > ### Author Response · Authors · 2025-08-08
> > > **Thanks for your positive feedback and score increase**
> > >
> > > Dear Reviewer UspK,
> > >
> > > We are delighted that our rebuttal has addressed your previous concerns and received your positive feedback!
> > >
> > > We greatly appreciate your recognition and the raised score.
> > >
> > > Once again, thank you for your time and valuable contributions.

---

### Official Review · Reviewer_CCcq · 2025-07-02

**Clarity:** 3
**Significance:** 3
**Originality:** 3
**Rating:** 5
**Confidence:** 4

**Summary:**

This paper identifies two limitations in existing contrastive learning frameworks—semantic divergence and unlearnability, and proposes SRA-CL to address them. The proposed method harnesses the semantic reasoning capabilities of LLMs to mitigate semantic divergence while introducing a learnable sample synthesizer that dynamically optimizes contrastive sample generation during training. Extensive experiments validate SRA-CL’s effectiveness and demonstrate its robust generalization across diverse recommendation models. Overall, this paper presents a well-motivated approach supported by comprehensive experimental validation.

**Questions:**

•	Could the authors explain why the Learnable Sample Synthesis is only applied to user-level contrastive learning but not extended to the item-level?
•	Could the authors analyze the training/inference efficiency of the proposed method, as well as the cost for API calls?
•	Why is cosine similarity preferred over dot product for semantic retrieval?

**Ethical Concerns:**

["NO or VERY MINOR ethics concerns only"]

**Final Justification:**

I have read the author's responses to my concerns in their rebuttal. Thank you for the efforts of the authors and the AC, which have helped me further reflect on the score evaluation.

**Quality:**

3

**Strengths And Weaknesses:**

Strengths:
•	Semantic divergence is a key challenge in contrastive learning for sequential recommendation. This work effectively addresses this issue by utilizing LLMs' semantic understanding and reasoning capabilities to retrieve semantic-consistent users/items, which is a well-motivated solution.
•	This paper proposes a learnable sample synthesizer to optimize contrastive sample generation during recommendation model training. This trainable approach helps ensure that the synthesized samples effectively enhance end-task performance.
•	This paper includes thorough experiments comparing different types of baselines and provides detailed ablation analysis. In addition, the generalization tests across various recommendation models and LLMs demonstrate the method's model-agnostic characteristic.
Weaknesses:
•	The paper does not discuss the training efficiency of the proposed method, despite its high inference efficiency. In addition, it doesn't provide cost analysis for API calls.
•	The Learnable Sample Synthesis proposed in this paper is only applied to user-level contrastive learning and not utilized at the item-level, with no specific explanation provided.

---

> ### Author Rebuttal · Authors · 2025-07-31
>
> We sincerely appreciate the reviewer's valuable time and insightful comments. Below are our point-by-point responses to the questions raised. We look forward to any additional feedback the reviewer may provide.
>
>
> # Response to W1 & Q2
> We sincerely thank the reviewer for recognizing the high inference efficiency of our method. In the main text, we have discussed inference efficiency, and we have included the training and inference algorithm workflows in Algorithm 1 and Algorithm 2 in the appendix. Here, we provide a detailed explanation of the inference and training costs.
> - **Inference Efficiency**: During inference, only the recommendation backbone is utilized. The contrastive learning tasks and the LLMs’ semantic embeddings are not involved in the inference process. This ensures that our framework can be deployed in real-world applications without incurring any additional inference latency from incorporating LLMs.
> - **Training Efficiency & Cost**: The training process of our method consists of two stages:
> In the first stage, we use an LLM API to obtain semantic information and convert it into embeddings, which are then cached to construct contrastive sample indices. The primary time cost in this stage comes from the API calls. However, by employing asynchronous concurrency, this step can be completed within a few hours. Crucially, this stage is performed once and requires no repetition during model training. Regarding the monetary cost of API usage, if using the DeepSeek API, all the experiments in this work can be covered within a budget of $50. Alternatively, if using an open-source model like Qwen-32B, this cost is reduced to zero, as it can be run locally without incurring API fees.
> In the second stage, we use the pre-constructed contrastive sample index to train the recommendation model. Regarding the computational complexity of this stage, our method maintains comparable time complexity to general ID-based contrastive recommendation approaches. The only additional overhead during training compared to conventional contrastive recommendation models comes from the learnable sample synthesis module—a single attention layer—whose parameter size is negligible compared to that of the main recommendation model.
>
>
> # Response to W2 & Q1
> - We sincerely apologize for any confusion caused. In fact, we have already addressed this question in the original paper. Specifically, Lines 202–208 on Page 6 clarify why the Learnable Sample Synthesis is applied only to user-level contrastive learning and not extended to the item level. Due to space constraints, we provided further elaboration in Appendix C.1, along with experimental comparisons in Table 4 of the appendix.
> - To reiterate, our preliminary experiments explored employing learnable synthesizers at the item level (analogous to inter-user contrastive learning) to generate substitute items. However, this approach yielded no measurable performance improvements (see Table 4, Appendix). We attribute this outcome to the inherent nature of item semantics, which are more straightforward to interpret and quantify than user preferences. Consequently, directly retrieving substitutes from semantically similar candidate pools proves both simpler and more reliable than attempting to match users with analogous preference patterns.
>
>
> # Response to Q3
> We appreciate the insightful question regarding our choice of cosine similarity for semantic retrieval. Our decision was based on the following considerations:
> - **Normalization robustness**: Unlike the dot product, cosine similarity inherently normalizes similarity scores by accounting for vector magnitudes. This property makes it more robust to scale inconsistencies in semantic embeddings, particularly since we do not apply explicit normalization to our embeddings.
> - **Alignment with the text embedding model’s pretraining objective**: The text embedding model we employed, SimCSE-RoBERTa, was explicitly trained using cosine similarity as its semantic similarity metric. By adopting the same measure for retrieval, we ensure consistency with the model’s original training objective, thereby enhancing the reliability of user similarity computations.

---

> > ### Comment · Reviewer_CCcq · 2025-08-05
> >
> > Thanks for the authors' detailed response, which has effectively addressed my concerns regarding efficiency and model specifics.  I think this paper successfully tackles a valuable problem in contrastive learning for sequential recommendation. So I prefer to maintain my score.

---

> > > ### Author Response · Authors · 2025-08-05
> > > **Thanks for Your Time and Positive Recognition**
> > >
> > > Dear Reviewer CCcq,
> > >
> > > we are very pleased to have addressed all your concerns. Once again, we sincerely appreciate your recognition and positive feedback on our work!

---

### Official Review · Reviewer_6Nm1 · 2025-07-02

**Clarity:** 3
**Significance:** 2
**Originality:** 3
**Rating:** 4
**Confidence:** 4

**Summary:**

This paper proposes Semantic Retrieval Augmented Contrastive Learning (SRA-CL), which aims to address the challenge of data sparsity in recommendation systems. By leveraging large language models (LLMs) to generate semantic embeddings of user preferences and items, the authors enhance the reliability of contrastive learning samples, thereby improving the accuracy and robustness of the recommendation outcomes. A key innovation of the proposed approach lies in the utilization of LLMs as semantic retrieval engines to augment the contrastive learning process. Furthermore, experiments on public datasets against baselines show significant improvements in HR@20 and NDCG@20. Ablation studies and hyperparameter analyses further validate the effectiveness and LLM-agnostic nature of each module.

**Questions:**

See Weaknesses

**Ethical Concerns:**

["NO or VERY MINOR ethics concerns only"]

**Final Justification:**

During the rebuttal, the authors provided further clarification on the LLM model selection,  application scenarios, and conclusion section, which partially addressed my concerns. Since these points do not significantly affect the main quality of the paper, I will maintain my original rating.

**Limitations:**

yes

**Quality:**

2

**Strengths And Weaknesses:**

Strengths:

This paper addresses a critical data sparsity challenge in recommender systems by identifying higher-quality positive behavior sequences for contrastive learning, thereby strengthening sequence modeling capabilities. The technical exposition is thorough and logically structured, and the authors back up their claims with extensive experiments and in-depth analyses. Moreover, the innovative use of LLMs to uncover semantically rich representations offers a promising new direction for further research and real-world application.

Weaknesses:

1. Both Qwen2.5-32B-Instruct and DeepSeek-V3 backbones yield similar HR and NDCG scores. Have the authors explored smaller versions of LLMs (e.g., 3B or 0.5B parameters)? Smaller models could significantly reduce inference latency and cost, making the approach more practical at scale.
2. Would directly encoding user semantics and collaborative signals using a pre-trained SimCSE-RoBERTa model produce comparable results to those of LLM-generated text outputs? A comparison could help clarify the added value of incorporating the LLM step.
3. Has the method been evaluated in scenarios where item or user text metadata is sparse or even completely omitted during LLM encoding?
4. The conclusion is relatively brief. It would benefit from a more detailed summary of the findings and a dedicated discussion of potential future work.

---

> ### Author Rebuttal · Authors · 2025-07-31
>
> We sincerely appreciate the reviewer's valuable time and insightful comments. Below are our point-by-point responses to the questions raised. We look forward to any additional feedback the reviewer may provide.
>
>
> # Response to Weakness 1
> We sincerely appreciate the reviewer’s valuable suggestion. Below, we answer this question in two aspects:
> - Our method does not rely on LLM inference during the prediction phase; instead, it only involves the inference of the backbone recommendation model. Therefore, our framework can be deployed in real-world large-scale applications without introducing additional latency from LLM integration. This also explains why our initial experiments did not explore smaller LLMs.
> - We fully acknowledge the reviewer’s insightful suggestion regarding smaller LLMs, as they could reduce training time and computational costs. Accordingly, we have conducted additional experiments using Llama3.2-3B-Instruct, and the results are presented below.
>
> |             |   Yelp  |         | Sports |         | Beauty |         | Office |         |
> |-------------|:-------:|:-------:|:------:|:-------:|:------:|:-------:|:------:|:-------:|
> |             | HR@20   | NDCG@20 | HR@20  | NDCG@20 | HR@20  | NDCG@20 | HR@20  | NDCG@20 |
> | DeepSeek-V3 | 0.1282  | 0.0533  | 0.0823 | 0.0347  | 0.1314 | 0.0568  | 0.1702 | 0.0725  |
> | Qwen2.5-32B | 0.1275  | 0.0526  | 0.0815 | 0.0338  | 0.1306 | 0.0559  | 0.1701 | 0.0720  |
> | Llama3.2-3B | 0.1258  | 0.0510  | 0.0801 | 0.0322  | 0.1288 | 0.0545  | 0.1680 | 0.0708  |
>
> The experimental results indicate that the 3B LLM underperforms compared to larger models. We attribute this to the fact that reasoning and summarizing user preferences require advanced inferential and abstraction capabilities, which smaller models tend to exhibit to a lesser degree than their larger counterparts. Consequently, the quality of generated user preference descriptions may be comparatively lower. In contrast, Qwen-32B, despite having fewer parameters than DeepSeek-V3, still maintains a sufficiently large scale to preserve most of these capabilities, thus showing less significant performance degradation. These observations suggest a potential trade-off between model scale and performance - smaller models may offer improved training efficiency, but correlate with diminished effectiveness in downstream tasks. We will include the experimental results and analysis in the appendix of the final version.
>
>
> # Response to Weakness 2
> - In fact, we had already conducted the experiment mentioned by the reviewer, and the results are presented in Table 2 of the Ablation Study section in the original paper. The "w/o LLM" setting denotes the scenario where the LLM is not used for text comprehension and processing—that is, the SimCSE-RoBERTa model directly encodes user and item texts, as the reviewer proposed.
> - The results in Table 2 indicates directly applying the text embedding model without LLM-based processing leads to performance degradation. This confirms that leveraging the LLM’s ability to understand and reason about user preferences is crucial. Without secondary processing by the LLM, the textual data from user historical behaviors becomes excessively verbose, with key information obscured by irrelevant details. Such lengthy and noisy text poses significant challenges for the embedding model, resulting in lower-quality embeddings. The LLM’s reasoning and summarization capabilities effectively mitigate this issue.
>
>
> # Response to Weakness 3
> We sincerely thank the reviewer for their insightful question and suggestion. The primary application scenario of this study focuses on cases where textual attributes of items are available, consistent with prior work [1, 2]. The metadata in the datasets used in our experiments exhibits a certain degree of sparsity, typically below 20%. We noticed that most real-world applications, such as Amazon, Taobao, Netflix and YouTube, generally provide and maintain high-quality textual metadata. For scenarios where text metadata is extremely sparse or entirely unavailable, such cases pose significant challenges for all LLM-based recommendation methods. However, addressing these challenges is beyond the scope of this paper due to space and time constraints. We plan to explore this direction further in future work. Once again, we deeply appreciate the reviewer's valuable suggestion.
>
> [1] Ren, Xubin, et al. Representation learning with large language models for recommendation. WWW' 24
>
> [2] Liu, Qidong, et al. Llm-esr: Large language models enhancement for long-tailed sequential recommendation. NeurIPS' 24
>
>
> # Response to Weakness 4
> We sincerely thank the reviewer for their suggestion regarding the Conclusion section. Your feedback will help improve our paper further. In the final version, we will expand the Conclusion and include a discussion on future work. The revised Conclusion is as follows:
>
> "*In this paper, we analyze the limitations of contrastive learning in sequential recommendation, namely Semantic Divergence and Unlearnability. To address these issues, we propose SRA-CL, a novel framework that enhances contrastive sample construction by integrating LLM-based semantic retrieval with a learnable sample synthesizer. SRA-CL leverages the capabilities of LLMs without increasing the inference time of the recommendation model, making it practical for large-scale real-world applications. Through comprehensive experiments, we demonstrate that LLM-based semantic-guided contrastive sample construction improves the contrastive learning, and we validate the effectiveness of the learnable sample synthesis mechanism. Furthermore, experiments with different recommendation model backbones confirm the model-agnostic nature of our approach. Looking forward, there are potential directions for future work: (1) optimizing the computational overhead of LLMs during the training process, and (2) exploring how to adapt our method to scenarios with scarce textual information.*"

---

> > ### Comment · Reviewer_6Nm1 · 2025-08-04
> >
> > Thanks for the clarification, which addresses some of my concerns. I will keep my scores.

---

> > > ### Author Response · Authors · 2025-08-04
> > > **Thanks for Your Time and Positive Recognition**
> > >
> > > Dear Reviewer 6Nm1,
> > >
> > > We are pleased that our rebuttal has addressed some of your concerns, and we would be happy to provide more information if you have any further questions.
> > >
> > > We sincerely appreciate your positive recognition of our work once again!

---

### Official Review · Reviewer_rXdN · 2025-07-02

**Clarity:** 3
**Significance:** 2
**Originality:** 2
**Rating:** 2
**Confidence:** 3

**Summary:**

This paper proposes Semantic Retrieval Augmented Contrastive Learning (SRA-CL) to enhance the performance of classical sequential recommendation models. Specifically, SRA-CL explores inter-user contrastive learning as well intra-user contrastive learning paradigms, where positive and negative pairs are constructed according to semantic similarity derived by LLM. The proposed framework is model-agnostic and can be applied to a lot of sequential recommendation models using contrastive learning. Enhanced by SRA-CL, the performance of sequential recommendation models improves by ~10% compared with existing sequential recommendations models aided by contrastive learning.

**Questions:**

Please refer to the weakness I listed above.

**Ethical Concerns:**

["NO or VERY MINOR ethics concerns only"]

**Limitations:**

Limitations are discussed. One other limitation could be applicability to large-scale data.

**Paper Formatting Concerns:**

No formatting concern.

**Quality:**

2

**Strengths And Weaknesses:**

Strength:
1. I think the question this paper tries to study is valid and the proposed framework seems like a reasonable approach to solve this problem (i.e., improving the quality of negative/positive pairs in contrastive learning for sequential recommendation).
2. the paper is nicely written and easy to follow.

Weakness:
1. While the question is valid, as a practitioner in sequential recommendation I don't find it to be very interesting as the proposal doesn't really spur too much intellectual thinking. I feel like this paper is better positioned at an IR conference rather than NeurIPS that emphasizes a lot on understanding/learning.
2. Efficiency and costs should be discussed -- I think it is trivial to leverage LLM to improve the performance of classical recommendation models. What is the trade-off for using LLMs to help recsys.
3. Following up on 2, there are a lot of new methods under the umbrella of generative recommendation, another totally different horizontal of using LLMs for sequential recommendation. How does the proposal compare against those (e.g., performance-wise, efficiency-wise, cost-wise, deployment difficulty-wise, etc).

---

> ### Author Rebuttal · Authors · 2025-07-31
>
> We sincerely appreciate the reviewer's valuable time and insightful comments. Below are our point-by-point responses to the questions raised. We look forward to any additional feedback the reviewer may provide.
>
> # Response to W1: About the Contribution and Positioning
> ### **1) The Contribution and Significance**:
> - Our work provides an in-depth analysis and novel exploration of contrastive learning in recommender systems. Unlike CV or NLP, contrastive learning in recommender systems presents unique challenges that render direct adoption of methods from other domains ineffective.
> - Furthermore, while existing contrastive learning approaches in sequential recommendation have predominantly relied on ID-based representations, we critically examine their limitations and, for the first time, propose leveraging semantic information to construct higher-quality learnable contrastive pairs, which is achieved by harnessing the understanding capabilities of LLMs.
> - We believe our work offers a significant contribution by redefining the understanding and application of contrastive learning in recommender systems. Our approach represents a meaningful and successful innovation, as mentioned by Reviewer 6Nm1: "The innovative use of LLMs to uncover semantically rich representations offers a promising new direction for further research and real-world application."
>
> ### **2) On the Positioning**:
> - Our paper aligns with NeurIPS's scope. Regarding the comment of “*this paper is better positioned at an IR conference rather than NeurIPS*”, we respectfully clarify that while specialized domains have dedicated venues (e.g., CVPR for computer vision, ACL for NLP, SIGIR for IR), NeurIPS serves as a premier interdisciplinary platform that explicitly welcomes cross-domain contributions—including research bridging information retrieval and machine learning.
> - Critically, our work focuses on user/item **understanding** and contrastive **learning**—precisely the areas highlighted by the reviewer as "***NeurIPS's emphasis on understanding/learning.***"
> - Moreover, the acceptance of **21 papers** at **NeurIPS 2024** featuring the terms "**recommendation**" or "**recommender**" in their titles further validates the suitability of our work for this venue. Representative examples include:
>   - Q. Liu et al. "LLM-ESR: Large Language Models Enhancement for Long-tailed Sequential Recommendation"
>   - Y. Liu et al. "End-to-End Learnable Clustering for Intent Learning in Recommendation"
>
> # Response to W2: About the Efficiency
> We appreciate the reviewer’s attention to efficiency. Actually, we have clarified the inference efficiency in Section 3.3 (Lines 220–224 of Page 6).
> - **Inference Efficiency**: During inference, only the lightweight backbone recommendation model is executed, while contrastive learning tasks and semantic embeddings are entirely disabled. This design ensures no additional computational overhead from LLM integration, preserving the same inference latency as the original recommendation backbone. Thus, our approach remains deployment-friendly and scalable for real-world systems, as it introduces no runtime cost to the production pipeline.
> - **Training Efficiency & Cost**: The training process of our method consists of two stages:
> In the first stage, we use an LLM API to obtain semantic information and convert it into embeddings, which are then cached to construct contrastive sample indices. The primary time cost in this stage comes from the API calls. However, by employing asynchronous concurrency, this step can be completed within a few hours. Crucially, this stage is performed once and requires no repetition during model training. Regarding the monetary cost of API usage, if using the DeepSeek API, all the experiments in this work can be covered within a budget of $50. Alternatively, if using an open-source model like Qwen-32B, this cost is reduced to zero, as it can be run locally without incurring API fees.
> In the second stage, we use the pre-constructed contrastive sample index to train the recommendation model. Regarding the computational complexity of this stage, our method maintains comparable time complexity to general ID-based contrastive recommendation approaches. The only additional overhead during training compared to conventional contrastive recommendation models comes from the learnable sample synthesis module—an adapter and a single attention layer—whose parameter size is negligible compared to that of the main recommendation model. Specifically, each experiment was conducted on a single 32GB V100 GPU and completed within 3 hours of training.
>
> # Response to W3: Discussion of LLM-based Methods
> Our paper has provided a comprehensive comparison and discussion of state-of-the-art LLM-based sequential recommendation methods. This is demonstrated through:
> 1) **Experiments**: Our experimental design includes three SOTA LLM-based sequential recommendation methods as baselines, all of which were recently published in top-tier conferences such as NeurIPS and SIGIR. Additionally, we have compared against seven contrastive learning approaches and three classical recommendation methods, resulting in a total of 13 baseline methods. We believe this constitutes a thorough comparative studies in this domain as the primary objective of our work is to enhance existing contrastive learning methods in sequential recommendation using LLMs.
> 2) **Related Work**: In the related work section (Section 5), we have systematically reviewed the latest advancements in the topic of LLMs for sequential recommendation and clearly delineated the distinctions between our work and these existing approaches.
>
> Below is the summarization of the comparison in terms of performance, efficiency, cost, and deployment difficulty:
> - **Performance**: Our approach demonstrates consistently better recommendation accuracy across multiple benchmarks, validating its effectiveness.
> - **Computational Efficiency and Cost**: 1) Inference Efficiency: Our method eliminates the need for LLM involvement or semantic processing during inference, making it as efficient as traditional recommendation models. 2) Training Cost: While leveraging semantic information, our framework maintains comparable training costs to other LLM-based methods.
> - **Scalability & Deployability**: Due to its lightweight inference mechanism, our approach is highly scalable and readily applicable to real-world large-scale recommendation systems.
>
>
> # Response to Reviewer's Comment Regarding Large-scale Data Applicability
> Our method is **deployment-friendly** and **scalable for real-world applications**, as it introduces no additional inference latency from incorporating LLMs. As discussed in Line 220-224 of Page 6, during inference, only the recommendation backbone is utilized. The contrastive learning tasks and LLMs’ semantic embeddings are not involved in the inference process. This design ensures no additional computational overhead from LLM integration, preserving the same inference latency as the original recommendation backbone.

---

> ### Author Response · Authors · 2025-08-06
>
> Dear Reviewer rXdN,
>
> We sincerely appreciate the time and effort you've dedicated to reviewing our work. We have carefully addressed all your comments in our rebuttal regarding:
> - Contribution and Positioning,
> - Efficiency,
> - Discussion of LLM-based Recommendation Methods
>
> Please let us know if these responses have adequately addressed your concerns.
>
> Thank you again for your valuable insights. We look forward to your reply.

---

### Note · Authors · 2025-08-12

Dear SACs, ACs, and Reviewers,

Thank you for your time and constructive feedback on our work. We deeply appreciate the reviewers’ positive remarks on various aspects of our paper, such as the **significance of the research question** (**all four reviewers**), **the novelty of our method and the thorough experimental validation** (**Reviewer 6Nm1, CCcq, and UspK**).

At the same time, we are delighted that **our rebuttal successfully addressed the concerns of three reviewers engaged in the discussion (6Nm1, CCcq, and UspK) and received their positive recognitions**, including one reviewer (UspK) raising their score.

However, we regret to note that **Reviewer rXdN did not respond to our rebuttal or engage in discussion**, despite we have addressed those concerns in our rebuttal.

Below are our final remarks regarding **Reviewer rXdN**’s concerns, and more details can be found in our rebuttal:

1.  **Conference Positioning**. Regarding the comment "*This paper is better positioned at an IR conference rather than NeurIPS*", we respectfully disagree for the following reasons: **1) Alignment with NeurIPS Criteria**: Our work explicitly advances contrastive learning by deepening the understanding of user/item semantics, which directly addresses the reviewer’s own emphasis on "understanding/learning." **2) Contribution**: Our work offers a significant contribution by redefining the understanding and application of contrastive learning. Three reviewers highlight the novelty and inspiration of our method. **3) Precedent at NeurIPS**: The conference regularly publishes work in recommender systems.

2.  **Efficiency and Deployability**. **1) Inference**: As clearly stated in our paper, **our method introduces no additional computational overhead from LLM integration during inference, maintaining identical latency to the original recommendation backbone. Thus, our method is deployment-friendly and scalable for real-world system**. **2) Training**: Our approach achieves comparable training efficiency to conventional LLM-based recommendation methods. **We note that Reviewer CCcq has endorsed our efficiency-related responses**.

3. **Discussion of LLM-based methods**. **Our paper has provided a comprehensive comparison of SOTA LLM-based methods in *Section 4. Experiments* and *Section 5. Related Work***.

We sincerely hope the reviewers and ACs will take our detailed clarifications into full consideration during final evaluation.

Thanks again.

Authors of Paper #3438

---

### Decision · Program_Chairs · 2025-09-17

**Decision:**

Accept (poster)

**Comment:**

The paper presents an LLM-based approach to improve contrastive learning for sequential recommender systems. The proposed framework utilizes LLM as a semantic retrieval engine to augment the contrastive learning process and achieves better performance in practice. Ablation studies are also sufficient.

Most of the concerns raised by the reviewers are well addressed in the rebuttal, including efficiency and results with smaller LLMs. The only remaining concern is about the venue fit (raised by Reviewer rXdN). However, the AC thinks that developing an improved algorithm for sequential recommendation is a good fit for NeurIPS, so we recommend acceptance for this paper.